# Characterisation of the ABO Blood Group Phenotypes Using Third-Generation Sequencing

**DOI:** 10.3390/ijms26125443

**Published:** 2025-06-06

**Authors:** Fredrick M. Mobegi, Samuel Bruce, Naser El-Lagta, Felipe Ayora, Benedict M. Matern, Mathijs Groeneweg, Lloyd J. D’Orsogna, Dianne De Santis

**Affiliations:** 1Department of Clinical Immunology, PathWest Laboratory Medicine WA, Fiona Stanley Hospital Network, Murdoch, WA 6150, Australia; 2School of Biomedical Sciences, The University of Western Australia, Queen Elizabeth II Medical Centre, Nedlands, WA 6009, Australia; 3Research and Advanced Computing, BizData Pty Limited, Melbourne, VIC 3000, Australia; 4The Transplant Immunology Department, Maastricht University Medical Centre, 6229 HX Maastricht, The Netherlands; 5UWA Medical School, The University of Western Australia, Perth, WA 6000, Australia

**Keywords:** ABO blood types, ABO subtypes, ABO genotyping, ABO phenotyping, third-generation sequencing, single-nucleotide variants

## Abstract

Third-generation sequencing (TGS), also known as long-read sequencing, has become a promising tool in clinical and research laboratories because it delivers high-resolution results with unmatched throughput. Specialised immunohematology laboratories currently employ sequencing-based methods to characterise rare ABO blood group phenotypes that cannot be identified through serology and genotyping methods. However, routine clinical application of these methods remains elusive due to the absence of validated laboratory protocols and bioinformatics tools. In this study, we have developed and validated a TGS-based workflow for comprehensive determination of the clinically relevant ABO phenotypes from DNA isolated from buccal swabs or whole blood. The region spanning exons 2 to 7 of the *ABO* gene were amplified and sequenced on MinION 10.4.1 flow cells. Predicted ABO phenotypes were initially determined based on single-nucleotide variants at gDNA261 (rs8176719), gDNA796 (rs8176746), and gDNA803 (rs8176747). However, certain O subtypes lacked the distinguishing deletion (rs8176719) and instead exhibited variations in exon 7 at gDNA802 (rs41302905) and gDNA805, caused by gDNA804 (rs782782485), which differentiate them from A alleles sharing the same nucleotides at gDNA261, gDNA796, and gDNA803. These additional variants were added to the analysis pipeline to identify the additional subtypes. DNA sequence data were sufficient to distinguish between the four clinically relevant ABO blood group phenotypes based on five polymorphic positions. While high sequencing coverage allowed for higher resolution genetic analysis, as few as 20 reads are sufficient for determining the ABO genotype and predicted phenotype of an individual. Typing results generated by this pipeline showed remarkable concordance with both serological results and molecular typing results by an independent laboratory, indicating its accuracy and reliability. This study demonstrates a comprehensive characterisation of clinically relevant ABO blood genotypes and predicted phenotypes using TGS methods. The approach provided a scalable and precise method for routine ABO blood group screening and aided in the development of pioneering bioinformatics tools suitable for clinical and research application.

## 1. Introduction

The impact of the ABO blood group system on various medical contexts, including pregnancy, blood transfusion, and solid organ and stem cell transplantation cannot be overstated [1,2,3,4]. The ABO system classifies human blood into exclusive phenotypes of O, A, B and AB based on the presence or absence of specific A and B antigens on the surface of red blood cells. ABO blood group antigens, although commonly associated with red blood cells, are expressed on numerous human tissues, including epithelial and endothelial cells. Expression of these antigens is orchestrated by glycosyltransferases that catalyse the addition of sugar residues onto the precursor H antigen, which is essential in defining blood types [5].

The alpha 1-3-N-acetylgalactosaminyltransferase and alpha 1-3-galactosyltransferase (*ABO*) gene is located on a region spanning ~20 kb on chromosome 9 and is expressed as seven exons. Exons 6 and 7 encode the major proportion of the *ABO* gene, including the catalytic domain of the glycosyltransferases that mediate the expression of A and B antigens that form the characteristic antigenic structures of the ABO blood groups [6,7]. The locus is also highly polymorphic, resulting in considerable genetic diversity and blood group-related polymorphisms in the exons, introns, and upstream enhancer region [8]. Genomic sequencing technologies have facilitated the characterisation of numerous polymorphisms in the *ABO* gene. While these technologies have aided the identification of polymorphisms responsible for the clinically relevant ABO genotypes, including AA, BB, AB, OO, OA, and OB [5,9,10,11,12,13], some polymorphisms in the ABO blood group gene have been shown to only alter the activity of the enzyme rather than its specificity [8]. This results in weaker A and B blood group phenotypes complicating delineation of ABO subtype-causing variants in diverse populations [14].

Exons 6 and 7 harbour important single-nucleotide variants (SNVs) and indels (deletions, insertions, or duplications) which reliably distinguish the A, B, and O alleles [11]. The O phenotype is characterised by a deletion at gDNA261 (rs8176719) in exon 6 [9,15]. To distinguish between A and B phenotypes, polymorphisms at gDNA796 (rs8176746) and gDNA803 (rs8176747) in exon 7 are used as they are present on all common and most rare A and B phenotypes [16]. Although these variants are sufficient to classify most common alleles of the four basic phenotypes (O, A, B, and AB), certain rare subtypes of the O phenotype lack the distinguishing deletion in exon 6 (rs8176719). Instead, they harbour variations in exon 7 at gDNA802 (rs41302905) and gDNA805, caused by gDNA804 (rs782782485) that would differentiate O allele subtypes from A alleles sharing the same nucleotides at gDNA261, gDNA796, and gDNA803 [13,15].

Serological methods traditionally used to phenotype ABO blood groups have transformed recipient and donor profiling while significantly contributing to identification of over 300 ABO alleles observed across cells [17]. However, in cases like registry typing with buccal swabs where erythrocytes are unavailable, or when serological testing is inconclusive or impractical due to the lack of antisera, molecular genotyping is necessary. Developing an ABO blood group screening test that is performed from DNA extracted from buccal swabs could facilitate this process and make it achievable in registries globally.

While the use of Oxford Nanopore Technologies (ONT) MinION sequencing in blood group typing has been demonstrated [9], the lack of a simple-to-use bioinformatics pipeline capable of automatically interpreting the data and perform ABO genotyping hampers widespread adoption.

In this study, we have developed a comprehensive DNA amplification and sequencing protocol and created an end-to-end bioinformatics data analysis workflow for predicting ABO phenotypes when serological techniques cannot be utilised. This workflow was specifically designed for ABO blood group screening on donors recruited via buccal swab to the Stem Cell Donors Australia (SCDA) registry. The ABO blood group would then be confirmed by serology if a donor was selected for a patient undergoing haematopoietic cell therapy. The workflow utilises the power of third-generation sequencing on ONT MinION/GridION platforms to characterise single-nucleotide variants (SNVs) and indels (insertions/deletions) on the ABO locus [9,18,19]. The pipeline is developed on Nextflow [20], a workflow manager that makes it easier to create and deploy pipelines. It follows the nf-core bioinformatics community best-practice guidelines to ensure accuracy and reliability [21,22]. This pipeline is highly adaptable for integration into existing workflows and allows future inclusion of additional genes into the genotyping pool at minimal extra cost and protocol complexity. The end-to-end workflow was validated on buccal swabs kindly provided by the SCDA from a donor pool enriched with European, South Asian, and East Asian ancestry.

The ABO analysis pipeline is available on GitHub at https://github.com/fmobegi/abo-analysis (accessed on 26 May 2025).

## 2. Results and Discussion

### 2.1. Optimising Laboratory Assays

To efficiently introduce the ability to predict ABO blood group phenotyping from DNA extracted from buccal swabs or saliva into routine testing, ABO specific primers were incorporated into the existing multiplexed Human Leukocyte Antigen (HLA) assay, allowing simultaneous testing of ABO and HLA from the same sample. The ABO amplicon size was 6.7 kb (Figure 1), which was of similar size to that of the HLA-A (3.0 kb), HLA-B (4.5 kb), HLA-C (3.3 kb), and HLA-DQB1*02 (5.6 kb) amplicons in the proposed assay. The similar size of the amplicons ensured that the ABO amplicon did not preferentially amplify at the expense of HLA genes.

Three primers, one forward and two reverse (Table 1), described in prior studies by Goebel et al. [5] and Wu et al. [11] were combined into two primer sets for the amplification of ABO exons 2 to 7. These sets were initially evaluated by gel electrophoresis on a 0.8% agarose gel to assess their specificity and amplification intensity, followed by single-molecule sequencing. Testing indicated both primer combinations were specific and effective in amplifying the expected ABO amplicon (Figure 2). The ABO primer pair ABO_IN1_Forward_Goebel and ABO_3UTR_Reverse_WU showed moderately stronger amplification across the tested samples and thus was used for further assay optimisation. The prioritised ABO primers were incorporated into the existing multiplex assay and tested at various concentrations to achieve even amplicon representation. The thermocycling conditions remained constant during testing to ensure the HLA amplicons were not negatively affected during the ABO optimisation. After optimising the primer concentrations in the multiplexed assay, a final ABO primer concentration of 1 µM was sufficient to achieve a minimum read depth of 20× for ABO variant calling while minimally impacting the amplification balance of HLA genes.

### 2.2. DNA Sequencing and ABO Phenotyping

Long-read sequencing was performed on the ONT MinION/GridION devices using R10.4.1 flow cells. Raw sequencing signals in FAST5 or POD5 format are emitted to a designated storage location on the local server. Following sequencing, these files are uploaded to the Azure Cloud for base calling and processing by the hybrid HLA-ABO pipeline (Figure 3). The bioinformatics and programming dependencies essential for running the pipeline are provided in Table 2. Specific combinations and proportions of ABO blood group-associated polymorphisms are also summarised in Table 3.

Existing FASTQ files can be processed using the standalone process as detailed in the pipeline’s GitHub repository (https://github.com/fmobegi/abo-analysis (accessed on 26 May 2025)). In our Azure implementation, however, FASTQ files intended for ABO variant calling and phenotype prediction are processed through the ABO workflow following base calling. For each barcoded sample and each of the two exon references, the workflow generates several preliminary analysis files and logs. They include a whole-exon variant-calling file (ReadAlignmentSpreadsheet.csv), ABO specific summary of the relevant variants (ABOReadPolymorphisms.txt), phenotype summary (ABOPhenotype.txt), BAM alignments and associated metrics files, and analysis logging files (Figure 4A). The pipeline also generates detailed amalgamated ABO typing results files and associated logs (ABO_result.xlsx, ABO_result.txt, ABO_result.log), a summary file for exporting into an in-house database (final_export.csv), a software dependencies and versions file (versions.yml), pipeline execution timeline and tracing reports (execution_timeline.html, execution_report.html, execution_trace.txt, workflow.oncomplete.txt), and FastQC/MultiQC reports for all input files within a sequencing run (Figure 4B).

### 2.3. Orchestration and Benchmarking Against the Validation Criteria

The following acceptance criteria were applied to assess the suitability of this workflow for routine clinical application: scope and expectations, effective data management, data and internet security, scalability, flexibility, reproducibility, suitability of computational tools for data analysis, accuracy and concordance, and traceability and quality control.

Minimap2, a well-documented aligner for mapping long reads, is used together with custom scripts written in Python3 to process the datasets [23]. SAMtools [24], BCFtools [25], FastQC [26], and MultiQC [27] are used to provide standardised variant analysis and quality control (QC) metrics. These tools were encapsulated in lightweight containers to manage versions and dependencies and ensure that the software environment remains consistent and reproducible regardless of where the workflow is run. Pipelining of the analysis scripts was orchestrated using Nextflow [20], aiding in reproducibility and providing a platform for defining and executing the pipeline in a consistent manner [21]. Nextflow’s declarative and version-controlled approach ensures that the workflow can be easily shared, executed, and reproduced across different computing environments. Its inbuilt checkpoints and safeguards further ensure that analysis execution and timeline logs are provided to the user to aid in debugging and process enhancement. Reproducibility is ensured through comprehensive documentation and standardised protocols, enabling the consistent replication of results across different settings or laboratories. All quality control metrics and reports, including runtimes, logs from each tool used, and versions of all dependencies, are also systematically logged and made available for manual review alongside the pipeline results.

To ensure regular, efficient usage of computing resources on the Azure cloud computing environment, the workflow and containers are executed and managed autonomously by the Loome Agent (https://www.loomesoftware.com) as before [28]. Loome creates on-demand resources and orchestrates data analysis following the concepts of IaaS (Infrastructure as a Service), PaaS (Platform as a Service), and SaaS (Software as a Service).

To validate the assay and the pipeline, a total of 130 samples were sequenced (some in duplicate runs) and processed using the ABO phenotyping pipeline, in both Azure Test and Azure Production environments. Same input datasets processed repeatedly in the two environments produced consistently identical variant calling and predicted phenotyping results with similar execution timelines and quality control metrics within acceptable deviation margins (Table 4). These results reinforce the traceability and reproducibility across different compute environments.

Serological typing results were provided by the Stem Cell Donors Australia (SCDA), and a further 50 samples were typed using sequencing-based methods at an independent commercial laboratory (HistoGenetics LLC Medical laboratory, New York 10562, NY, United States). Accuracy was determined by concordance between serological typing and molecular typing results (in-house and HistoGenetics). For samples that yielded more than 20 sequencing reads (a theoretical minimum required to obtain reliable variant calls), 7 showed a discordant ABO phenotype between the in-house assay and expected serologically verified results (Table 5). By including two additional polymorphisms that differentiate the A and O2 subtypes, six of these discrepancies were resolved. The remaining discrepant sample was included in the confirmatory samples sent to HistoGenetics to help investigate discrepancies and corroborate the serologically typed validation set. Of the 50 samples independently genotyped at HistoGenetics, 2 returned no results (Table 6). The sample that was not concordant with serological results showed concordance for ABO genotype and phenotype between the in-house pipeline and HistoGenetics results. Further investigation of the sample processing history attributed the failure to a pre-analytical technical error occurring prior to receiving the sample in the laboratory. ABO genotyping results for a second sample were identified as discordant between the Histogenetics laboratory and the in-house assay. The sample was typed as “AA” using the in-house assay, while HistoGenetics typed the sample as “AO” (Table 6). The raw Illumina fastq data were requested from HistoGenetics and processed through the in-house pipeline. The results were identical to those supplied by HistoGenetics, indicating a possible sample switch during shipment. The ABO phenotyping results using our pipeline are detailed in the Appendix A. The ABO phenotype concordance was 100% after resolution of the two sample processing errors for those samples with ≥20× read depth.

To further assess the performance and accuracy of the ABO assay in a large cohort, DNA extracted from buccal swabs from 910 individuals (1164 total samples including repeated DNA extraction attempts) were sequenced and processed through the ABO phenotyping pipeline. Serological results for six of these samples were not available. The quality of DNA extracted from buccal swabs may not always be as robust as that obtained from peripheral blood and has been shown to result in higher failure rates due to increased issues of DNA fragmentation and fewer high-quality long DNA strands [29]. Of the 1164 results, 206 (17.7%) did not meet the read depth threshold of 20× and were excluded from concordance analyses. There were six discrepancies identified for the 958 results that achieved the minimum read depth of 20× (Appendix A). Three of the discrepant results showed discordant HLA genotyping results due to sample processing issues. The remaining three discrepant results produced an “Unknown” result in the molecular phenotyping report. Despite having the gDNA261 (rs8176719) deletion in exon 6, typical of the O1 subtype, these samples exhibited a combination of polymorphisms in exon 7 that did not abut with the differentiating rules summarised in Table 3. The “unknown” results were investigated manually and found to align accurately with reference sequences of known rare ABO alleles, indicating they were true variants rather than assay artefacts. The performance metrics of the ABO assay for the 910 individuals is presented in Appendix A. The pipeline demonstrated excellent performance for samples with ≥20 reads, achieving an overall accuracy of 99.07%. Blood groups A, B, and O were classified with near-perfect sensitivity, specificity, and precision (all ≥ 0.99), while the AB group also showed strong performance, with all corresponding metrics at or above 97%, highlighting the method’s suitability in accurately distinguishing major ABO blood types. The level of concordance at a threshold of ≥10, ≥20, and ≥30 reads was determined, and 100% concordance was achieved at all thresholds; however, at the lowest threshold of ≥10 reads, a higher number of manual corrections were required. Therefore, in line with the established threshold for HLA typing [30], a threshold of ≥20 reads was set. Of the 910 individuals, results were obtained on the first attempt in 724 (79.6%) of the individuals. Following a final attempt, 861 samples (94.6%) of the cohort passed the minimum read depth of 20× and provided ABO blood group results with 100% concordance. The assay failure rate was therefore 5.4%, with 3.1% of these failures due to unacceptable low sequencing coverage. Improvements to the failure rate could be achieved by increasing the sequencing data per sample or reducing the number of samples on the flow cell, thereby increasing the data per sample.

## 3. Materials and Methods

### 3.1. Primer Validation and Library Preparation

Our assay was optimised for ABO-HLA combined molecular typing from DNA extracted from either whole blood, dry buccal swabs (COPAN FLOQswabs^®^; COPAN Diagnostics Inc., Murrieta, CA, USA), or saliva (Oragene^®^ kits; DNA Genotek, Ottawa, ON, Canada). Primers designed by Wu et al. [11] and Goebel et al. [5] were used to amplify the region spanning exons 2 to 7 of the *ABO* gene. Existing HLA primers and DNA preparation protocols were used [28]. For ABO integration, the published primers were initially tested in isolation to ensure specificity to all four serological groups (A, B, AB, O) against a panel of DNA with known ABO genotypes.

The new combined HLA and *ABO* gene molecular assays used a multiplexed PCR method using four master-mixes: CI (HLA-A, -B, -C, -DQB1*02 and ABO), CIIa (HLA-DQB1), CIIb (HLA-DQA1, -DPA1, -DPB1, -DRB1*04/*07/*09/*10, -DRB5), and CIIc (HLA-DRB1, -DRB3, -DRB4). PCR products are visualised by 0.8% agarose gel electrophoresis, then purified with AMPure XP beads using a 0.6× ratio and normalised to 200 ng for library preparation. Sequencing libraries were prepared using ONT Native Barcoding Kit 96 V14 (SQK-NBD114.96; Oxford Nanopore Technologies, Oxford, UK). The DNA extraction, PCR, and sequencing assay hands-on time is approximately 8 h and can be performed over 2 days.

### 3.2. Sequencing and Base Calling

The DNA libraries were subjected to sequencing on the ONT GridION platform using ONT’s R10.4.1 MinION flow cell (FLO-MIN114; Oxford Nanopore Technologies, Oxford, UK). Base calling and demultiplexing were performed using the in-house base calling pipeline for data from the ONT MinION and GridION platforms [28]. Briefly, changes in ionic current as a DNA/RNA strand passes through protein pores during a sequencing run were recorded as raw time-series signal data in POD5 file format [31]. Raw files were progressively uploaded to Azure cloud storage as they were generated by the MinION/GridION device in an automated process [28]. Dorado v0.5.3, a GPU-optimised data processing toolkit provided by ONT, was used to process the raw POD5 files. First, the “*dorado basecalling*” command was used to convert the raw sequencing signals into nucleotide sequences stored in unmapped Binary Alignment Map (uBAM) format. The “*dorado demux*” and “*dorado summary*” commands were then used to, respectively, de-multiplex the base-called reads into FASTQ files based on distinct molecular barcodes and generate the sequencing metrics file for quality assessment. Finally, the FASTQ reads were filtered using NanoFilt [32] with a preset length and quality threshold [28]. The quality of sequencing reads from each sample was assessed using FastQC [26] and NanoPlot [33].

### 3.3. Reads Mapping and Variants Detection

High-quality sequencing reads in FASTQ format were aligned to exon 6 and exon 7 of the ABO*A1.01.01.1 nucleotide reference sequences (NCBI Reference Sequence: NG_006669.2) using minimap2 [23]. The aligned sequences were converted into Binary Alignment Map (BAM) format using SAMtools [24]. Known genetic variants associated to common ABO phenotypes and their exonic localisation are published on the international society of blood transfusion website [34]. Of interest are polymorphisms in exon 6 and exon 7, which have been shown to accurately and unambiguously distinguish the ABO blood group status [13,15,16]. Nucleotide substitutions and indels were quantified using the Python3 module Pysam, a wrapper around htslib [35], SAMtools, and BCFtools packages [24,25]. Pysam implements processing and manipulation of alignment and variant files in Python using bcftools pileup iteration over the aligned read-length relative to the reference sequence. It handles the Sequence Alignment Map (SAM) in compressed BAM or Compressed Reference-oriented Alignment Map (CRAM) format, and genetic variation information in Variant Call Format (VCF) or Binary Variant Call Format (BCF). For each individual sample, the pipeline logs quality metrics for raw and trimmed reads QC, read alignment to the reference exons, variant-calling, and variant-filtering steps. These are included in the standard output to aid troubleshooting.

### 3.4. ABO Variant Quantification, ABO Phenotyping, and Reporting

Proportions of single-nucleotide variants and indels for all aligned reads were pooled for each sample. Relevant polymorphisms were computed as a “match-mismatch” percentage to determine the ABO phenotypes.

In addition to the main ABO typing results, the pipeline generates various analysis logs to help with debugging. ABO typing results from each sample are collated into a simple Excel spreadsheet for each sequencing run. Preliminary analysis reports and logs for each sample are summarised using MultiQC [27]. The total data analysis time for a batch run of 24 samples is approximately 2-4 h.

Samples with <20× at relevant positions are flagged as “Low coverage” or “Unknown” in the “ABO_result.xlsx”, which is the main output of this workflow. These samples are excluded from reporting in the “final_export.csv” file and flagged for repeat testing or manual analysis if appropriate.

### 3.5. The Core Code, Pipelining, and Deployment

One of the main challenges of implementing sequencing-based methods for clinical testing is developing an ecosystem of information technology infrastructure for resilient, reproducible, and scalable genomics within an often low-throughput health-centric setup. Nextflow and container runtimes are instrumental in addressing these challenges [20,36]. Nextflow is a workflow management engine and domain-specific language (DSL) that makes it easy to create reproducible workflows that are interoperable across different local, cloud, and cluster computing infrastructures. It supports software containers and seamlessly integrates most common command-line and scripting languages, facilitating complex data manipulations. Installation and maintenance of bioinformatics software dependencies is a challenging task and a common source of irreproducibility in scientific pipelines. Container runtimes like Docker, Singularity, and Conda streamline this process by packaging required software dependencies into standardised, immutable, and ready-to-run units called containers [37]. These containers provide portable infrastructure-agnostic solutions for seamless deployment and maintenance of bioinformatics workflows across different systems or environments.

The base code for this workflow is split into 8 Python scripts, namely AnalyzeAbo.py, AnalyzeAbo_Main.py, AnalyzeAbo_Common.py, generate_samples_mapping.py, PairwiseAlignmentResult.py, BloodGroupStats.py, Aggregate_ABO_reports.py, and rename_samples.py. The primary analysis of sequencing reads is performed by the “AnalyzeAbo_Main.py” script, which imports functions from the other associated scripts. ABO analysis can be divided into 5 fundamental steps: aligning fastq reads to the reference, identifying variants, quantifying variants, resolving ABO phenotype, and generating results and QC reports.

These scripts are compatible with Python 3 (Python version ≥ 3.10). They are unified into a Nextflow workflow (compatible with Nextflow version ≥ 23) with the ability to containerise all core components, for each analysis step/process on the fly, based on a user-selected profile (Conda, Docker, Singularity). While this ABO phenotyping pipeline is deployable as a stand-alone solution, for our clinical testing purpose, the pipeline was deployed on the Azure cloud-computing service as a sub-workflow extending the functionality of an existing automated pipeline for HLA genotyping using ONT sequencing data [28].

## 4. Conclusions

The newly developed bioinformatics workflow for the predicted ABO phenotyping seamlessly integrates as a submodule into the existing HLA genotyping workflow, requiring minimal adjustments to both the core laboratory and computational processes, while introducing new functionality. This integration has optimised laboratory efficiency by providing an integrated platform for ABO blood group typing within an existing HLA typing workflow.

By utilising containerisation and an efficient workflow management engine, Nextflow, the pipeline can autonomously scale to efficiently utilise available computing resources and readily deploy on most Unix environments without modification. It also thoroughly follows WA Health’s cloud data management protocols for processing, reporting, storing, and managing TGS data, ensuring uniformity, accessibility, usability, safety, and trustworthiness. Furthermore, the pipeline effectively upholds stringent compliance requirements for health data governance by leveraging tailored and audited security solutions offered by Microsoft Azure Cloud Computing, as previously reported [28].

The ABO phenotyping workflow yielded accurate, reproducible, and concordant results, and was shown to be effective for routine clinical application and large-scale typing projects.

### Limitations and Future Directions

The ABO system has a high degree of polymorphism with numerous alleles, including many rare variants. While the algorithm’s fixed set of classification variant combinations and allele balance threshold (80:20) for heterozygous positions has been demonstrated to be robust and accurate, some SNVs failing this cutoff resulted in samples being classified as “unknown”, an issue that is more common with low sequencing depth. In these cases, manual analysis, whereby the allele balance threshold is reduced, may result in an allele call. This approach is particularly useful for samples that truly carry two alleles at the relevant heterozygous positions used for classification of common ABO phenotypes. In addition, as observed in ABO typing based on three polymorphic positions, some O subtypes were mistyped, prompting the inclusion of two additional polymorphisms to resolve them. Where applicable, future iterations of this pipeline will attempt to incorporate additional variants to further enhance typing accuracy within this primary context. However, there is the possibility of non-functional variants that are not easily detectable or functional variants associated with rare ABO alleles not being identified. There is no planned solution to address this issue in the current or future versions of the pipeline.

The pipeline is open-sourced allowing other users to contribute and improve its ability to distinguish additional ABO subtypes, especially A1 and A2, which are important for ABO-incompatible kidney transplants [38]. This effort is supported by the global nf-core bioinformatics community [22] to enhance timely maintenance and standardisation using peer-reviewed bioinformatics modules that support broader clinical utility. Development updates and contributions are accessible via the community repository at https://nf-co.re/abotyper/dev/ (accessed on 26 May 2025).

## Figures and Tables

**Figure 1 ijms-26-05443-f001:**
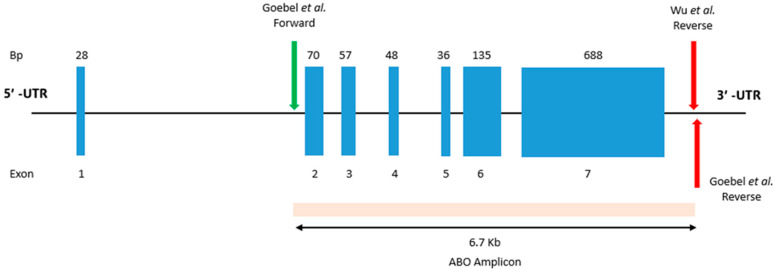
Visualisation of ABO amplicon. *ABO* gene coverage of a 6.7 kb amplicon, covering exons 2–7, produced from Geobel et al. [5] and Wu et al. [11] primers.

**Figure 2 ijms-26-05443-f002:**
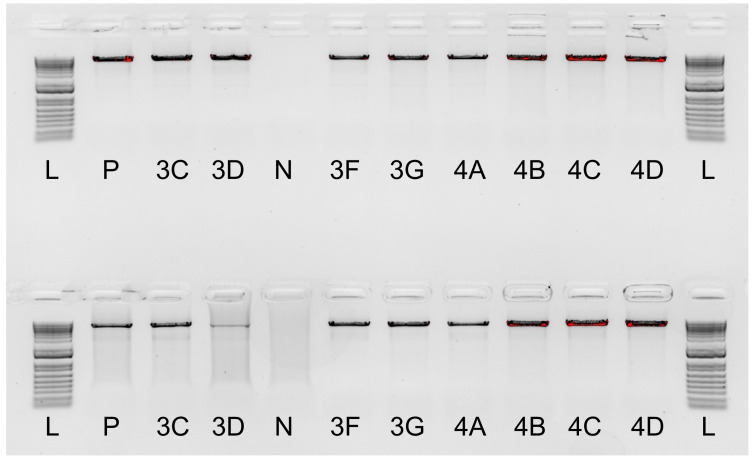
A gel electrophoresis image visualised on a 0.8% Agarose gel with 1 kb Plus DNA ladder. ABO primer pair testing was performed on positive control (P), negative control (N), and a panel of 8 samples compared to the ladder (L). The top panel represents amplification using ABO_IN1_Forward_Geobel and ABO_3UTR_Reverse_Geobel primer pair, while the bottom panel represents amplification with ABO_IN1_Forward_Geobel and ABO_3UTR_Reverse_WU.

**Figure 3 ijms-26-05443-f003:**
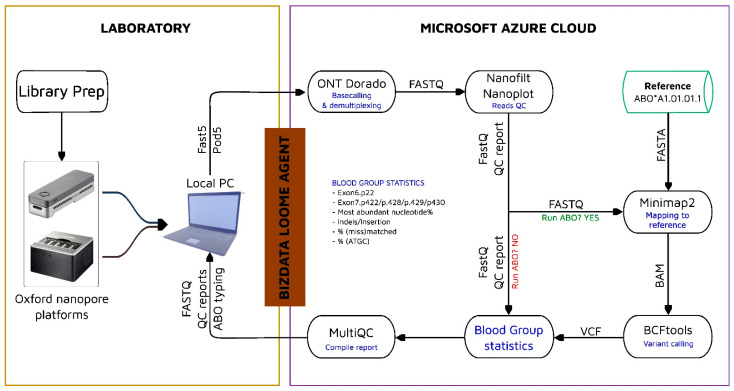
A schematic overview of the ABO genotyping pipeline. Raw sequencing signals generated by the ONT device in the laboratory are autonomously and gradually uploaded to Azure cloud for base calling, demultiplexing, quality filtering, and ABO genotyping. See detailed workflow in Appendix A.

**Figure 4 ijms-26-05443-f004:**
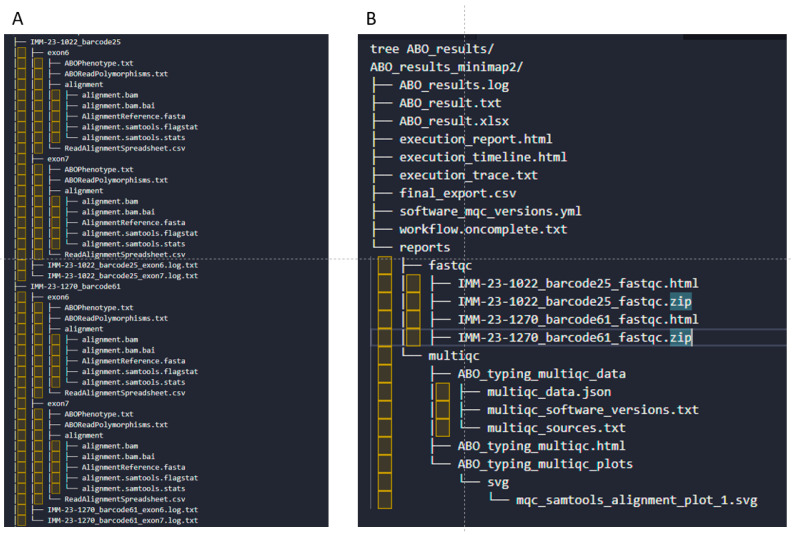
An overview of the ABO pipeline output files and directory structure. (**A**) Sample outputs per sample for each reference exonic sequence. (**B**) Pipeline execution log files and combined outputs for all samples.

**Table 1 ijms-26-05443-t001:** Primer sequences used in ABO genotyping assay development.

PrimerID *^a^*	Gene Location	Chromosome Location	Sequence (5′-3′)
ABO IN1 Forward Geobel	ABO Intron 1	9q34.2	GACCATCTTGGCAGATGAAGG
ABO 3UTR Reverse Geobel	ABO 3′ UTR	9q34.2	GCCTAGGCTTCAGTTACTCAC
ABO 3UTR Reverse Wu	ABO 3′ UTR	9q34.2	GGGCCTAGGCTTCAGTTACTC

*^a^* Primer identifier (PrimerID) includes their origin; “Goebel” denotes primers from Goebel et al. [5] and “Wu” denotes primers from Wu et al. [11].

**Table 2 ijms-26-05443-t002:** A list of bioinformatics tools and core dependencies required to run the ABO blood group typing pipeline. The minimum versions for each dependency are listed.

Purpose	Tool	Version
Reads QC	FastQC	0.12.1
Local alignment of FASTQ to reference	BWA	0.7.18
	Minimap2	2.28
Processing BAM files	BCFtools	1.21
	SAMtools	1.21
	htslib	1.21
Quantifying variants	Python + pip packages	3.10
	Bio	1.7.1
	BioPython	1.83
	openpyxl	3.1.5
	pandas	2.2.3
	pysam	0.22.1
	xlsxwriter	3.2.2
Reporting	MultiQC	1.25.1

**Table 3 ijms-26-05443-t003:** Polymorphisms in exon 6 and exon 7 and their derivative ABO phenotypes and genotypes. The coding sequence positions correspond with positions 22 on exon 6, and positions 422, 428, 429, and 431 on exon 7.

Reference SNP ID		rs8176719	rs8176746	rs41302905	rs8176747	rs782782485
Exon		6	7	7	7	7
Position exon (gene)		22(261)	422(796)	428(802)	429(803)	431(805)
Position GRCh38.p14		chr9:133257521-133257522	chr9:133255935	chr9:133255929	chr9:133255928	chr9:133255898
Amino acid change		p.Thr88Profs	p.Leu266Met	p.Gly268Arg	p.Gly268Ala/p.Gly268Val	p.Phe269Valfs*124
Nucleotide change		c.261delG/c.260_262insG	c.796C>A	c.802G>A/c.802G>C	c.803G>C/c.803G>T	c.804dupG
Consequence		frameshift	missense	missense	missense	frameshift
Phenotype	O1	Deletion	C	G	G	T
	O2	G	C	A	G	T
	O3	G	C	G	G	G
	A	G	C	G	G	T
	AB	G	C/A	G	G/C	T
	B	G	A	G	C	T
Genotype	O1A	Del 50%/G 50%	C 100%	G 100%	G 100%	T 100%
	O2A	G 100%	C 100%	A 50%/G 50%	G 100%	T 100%
	O3A	G 100%	C 100%	G 100%	G 100%	G 50%/T 50%
	O1B	Del 50%/G 50%	C 50%/A 50%	G 100%	G 50%/C 50%	T 100%
	O2B	G 100%	C 50%/A 50%	A 50%/G 50%	G 50%/C 50%	T 100%
	O3B	G 100%	C 50%/A 50%	G 100%	G 50%/C 50%	G 50%/T 50%
	O1O2	Del 50%/G 50%	C 100%	A 50%/G 50%	G 100%	T 100%
	O1O3	Del 50%/G 50%	C 100%	G 100%	G 100%	G 50%/T 50%
	O2O3	G 100%	C 100%	A 50%/G 50%	G 100%	G 50%/T 50%
	O1O1	Del 100%	C 100%	G 100%	G 100%	T 100%
	O2O2	G 100%	C 100%	A 100%	G 100%	T 100%
	O3O3	G 100%	C 100%	G 100%	G 100%	G 100%
	AA	G 100%	C 100%	G 100%	G 100%	T 100%
	BB	G 100%	A 100%	G 100%	C 100%	T 100%
	AB	G 100%	C 50%/A 50%	G 100%	G 50%/C 50%	T 100%

* Denotes a premature stop codon. Human Genome Variation Society (HGVS) nomenclature is used for nucleotide (c.) and protein (p.) changes. The 50:50 heterozygosity fraction is only for representation purposes. In practice, up to an 80:20 nucleotide split provided accurate classification and was considered for the development of this pipeline. In addition, the current pipeline may fail to classify some very rare ABO alleles based on the defined SNVs and therefore may require additional variants to be incorporated.

**Table 4 ijms-26-05443-t004:** ABO genotyping pipeline produces identical results across different deployment environments in TEST and PROD.

Sample	Azure Production Pipeline				Azure Test Pipeline				
	Reads	Phenotype	Genotype	Extended Genotype	Reads	Phenotype	Genotype	Extended Genotype	Reliability
Sample0040	8	O	OO	O1O1	8	O	OO	O1O1	Very Low (≤20 reads)
Sample0044	72	O	OO	O1O1	72	O	OO	O1O1	
Sample0048	16	O	OO	O1O1	6	O	OO	O1O1	Very Low (≤20 reads)
Sample0049	732	O	OO	O1O1	727	O	OO	O1O1	Robust (≥500 reads)
Sample0085	2678	O	OO	O1O1	2677	O	OO	O1O1	Robust (≥500 reads)
Sample0087	947	A	AO	AO1	943	A	AO	AO1	Robust (≥500 reads)
Sample0088	2547	B	BO	BO1	2539	B	BO	BO1	Robust (≥500 reads)
Sample0089	1298	O	OO	O1O1	1298	O	OO	O1O1	Robust (≥500 reads)
Sample0091	848	O	OO	O1O1	851	O	OO	O1O1	Robust (≥500 reads)
Sample0092	1448	O	OO	O1O1	1447	O	OO	O1O1	Robust (≥500 reads)
Sample0093	406	A	AO	AO1	405	A	AO	AO1	
Sample0097	3512	O	OO	O1O1	3521	O	OO	O1O1	Robust (≥500 reads)
Sample0098	1707	O	OO	O1O1	1712	O	OO	O1O1	Robust (≥500 reads)
Sample0099	239	O	OO	O1O1	240	O	OO	O1O1	
Sample0100	1852	A	AO	AO1	1854	A	AO	AO1	Robust (≥500 reads)
Sample0101	1203	O	OO	O1O2	1204	O	OO	O1O2	Robust (≥500 reads)
Sample0102	194	A	AO	AO1	193	A	AO	AO1	
Sample0103	651	O	OO	O1O1	651	O	OO	O1O1	Robust (≥500 reads)
Sample0105	1173	O	OO	O1O2	1172	O	OO	O1O2	Robust (≥500 reads)
Sample0106	35	B	BO	BO1	35	B	BO	BO1	Low (≤50 reads)
Sample0107	18	O	OO	O1O1	18	O	OO	O1O1	Very Low (≤20 reads)
Sample0110	1191	B	BO	BO1	1191	B	BO	BO1	Robust (≥500 reads)
Sample0111	26	O	OO	O1O1	26	O	OO	O1O1	Very Low (≤20 reads)
Sample0112	357	A	AO	AO1	358	A	AO	AO1	
Sample0113	943	A	AA	AA	945	A	AA	AA	Robust (≥500 reads)
Sample0114	314	O	OO	O1O1	313	O	OO	O1O1	
Sample0115	1286	O	OO	O1O1	1286	O	OO	O1O1	Robust (≥500 reads)
Sample0116	32	A	AO	AO1	32	A	AO	AO1	Low (≤50 reads)
Sample0117	13	O	OO	O1O1	13	O	OO	O1O1	Very Low (≤20 reads)
Sample0119	115	O	OO	O1O1	113	O	OO	O1O1	
Sample0120	1037	O	OO	O1O1	1039	O	OO	O1O1	Robust (≥500 reads)

**Table 5 ijms-26-05443-t005:** O allele subtypes deobfuscation using additional variants. Some rare subtypes of the O allele lack the distinguishing deletion in exon 6 but can be separated by the variants in nucleotide 802 and 805 of exon 7.

Sample	Reads	Exon 6 (pos 22)	Exon 7 (pos 422)	Exon 7 (pos 428)	Exon 7 (pos 429)	Exon 7 (pos 431)	3SNV Phenotype	3SNV Genotype	5SNV Phenotype	5SNV Genotype	Expected	Concordance
Sample0054	824	50%G/50%Del	100%C	50%G/50%A	100%G	100%T	A	AO	O	OO	O	Yes
Sample0061	1684	50%G/50%Del	100%C	50%G/50%A	100%G	100%T	A	AO	O	OO	O	Yes
Sample0071	432	100%G	50%C/50%A	50%G/50%A	50%G/50%C	100%T	AB	AB	B	OB	B	Yes
Sample0081	117	50%G/50%Del	100%C	50%G/50%A	100%G	100%T	A	AO	O	OO	O	Yes
Sample0101	575	50%G/50%Del	100%C	50%G/50%A	100%G	100%T	A	AO	O	OO	O	Yes
Sample0105	529	50%G/50%Del	100%C	50%G/50%A	100%G	100%T	A	AO	O	OO	O	Yes

**Table 6 ijms-26-05443-t006:** Concordance between sequencing-based and serological ABO typing results. Fifty samples were sequenced and genotyped in-house and at an independent commercial laboratory, and the results were akin to those genotyped serologically.

Sample	Histogenetics (Molecular)				In-House (Molecular)		Serology	Concordance *^a^*		
Sample0124	OO	O	-	O-	No result	No result	No result	No result	No result	Very Low (≤20 reads)
Sample0121	AO	A	+	A+	No result	No result	No result	No result	No result	Very Low (≤20 reads)
Sample0118	AO	A	+	A+	No result	No result	No result	No result	No result	Very Low (≤20 reads)
Sample0117	OO	O	-	O-	No result	No result	O	No result	No result	Very Low (≤20 reads)
Sample0107	OO	O	+	O+	No result	No result	O	No result	No result	Very Low (≤20 reads)
Sample0096	OO	O	+	O+	No result	No result	O	No result	No result	Very Low (≤20 reads)
Sample0090	AO	A	+	A+	No result	No result	A	No result	No result	Very Low (≤20 reads)
Sample0082	No result	No result	No result	No result	AA	A	A	No result	No result	Robust (≥500 reads)
Sample0080	No result	No result	No result	No result	OO	O	No result	No result	No result	
Sample1030	AA	A	+	A+	No result	No result	A	No result	No result	
Sample0113	AO	A	+	A+	AA	A	A	FALSE	TRUE	Robust (≥500 reads)
Sample0116	AO	A	+	A+	AO	A	No result	TRUE	TRUE	Low (≤50 reads)
Sample0115	OO	O	+	O+	OO	O	O	TRUE	TRUE	Robust (≥500 reads)
Sample0114	OO	O	-	O-	OO	O	O	TRUE	TRUE	
Sample0112	AO	A	+	A+	AO	A	A	TRUE	TRUE	
Sample0111	OO	O	-	O-	OO	O	O	TRUE	TRUE	
Sample0110	BO	B	+	B+	BO	B	A	TRUE	TRUE	Robust (≥500 reads)
Sample0109	OO	O	+	O+	OO	O	O	TRUE	TRUE	Low (≤50 reads)
Sample0108	AO	A	+	A+	AO	A	A	TRUE	TRUE	
Sample0104	OO	O	+	O+	OO	O	O	TRUE	TRUE	
Sample0103	OO	O	+	O+	OO	O	O	TRUE	TRUE	Robust (≥500 reads)
Sample0102	AO	A	+	A+	AO	A	No result	TRUE	TRUE	
Sample0101	OO	O	+	O+	OO	O	O	TRUE	TRUE	Robust (≥500 reads)
Sample0100	AO	A	+	A+	AO	A	A	TRUE	TRUE	Robust (≥500 reads)
Sample0099	OO	O	+	O+	OO	O	O	TRUE	TRUE	
Sample0098	OO	O	+	O+	OO	O	O	TRUE	TRUE	Robust (≥500 reads)
Sample0097	OO	O	-	O-	OO	O	O	TRUE	TRUE	Robust (≥500 reads)
Sample0095	OO	O	+	O+	OO	O	O	TRUE	TRUE	
Sample0094	OO	O	+	O+	OO	O	No result	TRUE	TRUE	
Sample0093	AO	A	+	A+	AO	A	No result	TRUE	TRUE	
Sample0092	OO	O	+	O+	OO	O	O	TRUE	TRUE	Robust (≥500 reads)
Sample0091	OO	O	+	O+	OO	O	O	TRUE	TRUE	Robust (≥500 reads)
Sample0089	OO	O	+	O+	OO	O	O	TRUE	TRUE	Robust (≥500 reads)
Sample0088	BO	B	+	B+	BO	B	B	TRUE	TRUE	Robust (≥500 reads)
Sample0087	AO	A	+	A+	AO	A	No result	TRUE	TRUE	Robust (≥500 reads)
Sample0086	OO	O	+	O+	OO	O	O	TRUE	TRUE	
Sample0085	OO	O	+	O+	OO	O	No result	TRUE	TRUE	Robust (≥500 reads)
Sample0084	OO	O	+	O+	OO	O	O	TRUE	TRUE	
Sample0083	AA	A	-	A-	AA	A	A	TRUE	TRUE	Robust (≥500 reads)
Sample0081	OO	O	+	O+	OO	O	O	TRUE	TRUE	
Sample0079	AO	A	+	A+	AO	A	A	TRUE	TRUE	
Sample0078	OO	O	+	O+	OO	O	O	TRUE	TRUE	
Sample0077	OO	O	+	O+	OO	O	O	TRUE	TRUE	
Sample0076	OO	O	+	O+	OO	O	O	TRUE	TRUE	Robust (≥500 reads)
Sample0075	OO	O	-	O-	OO	O	O	TRUE	TRUE	Low (≤50 reads)
Sample0074	OO	O	+	O+	OO	O	O	TRUE	TRUE	
Sample0061	OO	O	+	O+	OO	O	O	TRUE	TRUE	Robust (≥500 reads)
Sample0054	OO	O	-	O-	OO	O	O	TRUE	TRUE	Robust (≥500 reads)
Sample0020	OO	O	-	O-	OO	O	O	TRUE	TRUE	
Sample0018	BO	B	+	B+	BO	B	B	TRUE	TRUE	

*^a^* Concordance was determined between in-house and Histogenetics molecular results, and in-house molecular and serology results.

## Data Availability

The data supporting this study are included in the article and Appendix A. Raw sequencing data are unavailable due to ethical and privacy restrictions on sharing human genetic material.

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
