# Peer review of "Characterisation of the ABO Blood Group Phenotypes Using Third-Generation Sequencing"

_ijms, 2025, doi:10.3390/ijms26125443_

Round 1
Reviewer 1 Report
Comments and Suggestions for Authors
This manuscript describes the development of a sequencing-based assay to identify ABO blood type. This assay utilizes long read sequencing technology from Oxford Nanopore. This assay includes PCR, sequencing on an ONT sequencer, and bioinformatics. The bioinformatics portion of the assay is designed to function as a pipeline and is containerized such that it can be deployed in a consistent manner to yield reproducible results. The assay is designed to be added to an existing HLA genotyping protocol to yield additional valuable data with minimal additional effort.
Overall, the manuscript is well written.
When developing the assay, the authors began with three primers, one forward and two reverse, from other publications, which they tested to examine their suitability for this assay. In this description, it would be useful to say where they come from at the first description, rather than simply “three primers for amplification and sequencing…”. I was left wondering where these three primers came from until later in the manuscript.
Positions of SNVs are shown using two different coordinate systems. It was not always obvious which were being used. For example, in Table 5, the positions were 261, 796, etc. while the same SNVs in the supplementary table were 22, 422, etc. It is fine to initially describe both, but everywhere else, only one coordinate system should be used.
In the paragraph beginning at line 238, it is stated that some reads exhibited a combination of polymorphisms that do not fit the known patterns. Could they be PCR chimeras?
This method is designed to identify known alleles. For most samples, this is an efficient and effective approach. However, as noted in the introduction, there are over 300 known alleles of the ABO gene. It is possible, and perhaps likely, that there are other, yet unknown, alleles that will be discovered as the method is employed at a larger scale in a clinical environment. I would suggest adding functionality to the pipeline, perhaps hidden from the user under normal circumstances, that will identify when a novel allele is found and bring that to the user’s attention so that it may be added to the database. This is especially important when the novel variant is an indel which would result in a frameshift mutation and what would otherwise appear to be an A or B allele resulting in an O phenotype. This should at least be mentioned in the limitations and future directions section.
Despite the samples being normalized prior to pooling, there is a wide variation in the number of reads obtained in each sample. Approximately 20% of samples have fewer than the set lower limit of 20 reads and others have as many as several thousand reads. Is this due to uneven amplification of the amplicons in each pool? Is there any correlation with number of reads and allele that might indicate an allele with a SNV in the primer binding site? Did the samples with low numbers yield good HLA genotyping data? This failure rate was mentioned, but there was no discussion of why it was so high or any indication of effort to improve it. It will need to be improved prior to clinical use.
Author Response
|
Comments 1: When developing the assay, the authors began with three primers, one forward and two reverse, from other publications, which they tested to examine their suitability for this assay. In this description, it would be useful to say where they come from at the first description, rather than simply “three primers for amplification and sequencing…”. I was left wondering where these three primers came from until later in the manuscript. |
|
Response 1: While details about the primers are available in the first sentence of the M&M section, “Primers designed by Wu et al. [11] and Goebel et al. [5] were used to amplify region spanning exons 2 to 7 of the ABO gene. Existing HLA primers and DNA preparation protocols were used [28]. For ABO integration, the published primers were initially tested in isolation to ensure specificity to all four serological groups (A, B, AB, O) against a panel of DNA with known ABO genotypes.”, and also included in Figure 1 and in the “PrimerID” column of Table 1 which are referenced in text, this information may not be obvious to readers at first glance. To enhance clarity and provide proper attribution, we have revised the text at lines 119-123 of the manuscript as follows: "Three primers, one forward and two reverse (Table 1), described in prior studies by Goebel et al. [5] and Wu et al. [11] were combined into two primer sets for the amplification of ABO exons 2 to 7. These sets were initially evaluated by gel electrophoresis on a 0.8% agarose gel to assess their specificity and amplification intensity, followed by single-molecule sequencing." We have also added a table footnote “Primer identifiers (PrimerID) include their origin; "Goebel" denotes primers from Goebel et al. [5], and "Wu" denotes primers from Wu et al. [11].” to Table 1 at line 135-136 to provide additional information for the column “PrimerID and indicate the source of each primer for clarity. These revisions explicitly cite the sources of the primers at the first mention, addressing your concern about the lack of immediate attribution. |
|
Comments 2: Positions of SNVs are shown using two different coordinate systems. It was not always obvious which were being used. For example, in Table 5, the positions were 261, 796, etc. while the same SNVs in the supplementary table were 22, 422, etc. It is fine to initially describe both, but everywhere else, only one coordinate system should be used. |
|
Response 2: We acknowledge the presence of two coordinate systems in our manuscript—one in Table 5 and another in the supplementary tables. To ensure consistency, we have modified Table 5 to use the exonic positions, similar to the supplementary tables, as these represent the positions utilised in the pipeline output files. |
Comments 3: In the paragraph beginning at line 238, it is stated that some reads exhibited a combination of polymorphisms that do not fit the known patterns. Could they be PCR chimeras?
Response 3: We appreciate your insightful comment regarding the potential for PCR chimeras in the discrepant samples. As stated in the manuscript, these discrepancies are due to polymorphisms in exons 6 and 7 which do not fulfill the differentiating rules (for the main ABO phenotypes) summarised in Table 3. The observed patterns correspond to known but rare ABO alleles that we have not included in our classification algorithm at this time but can be resolved using manual interpretation by the testing scientist when required.
To reinforce this, we have updated the manuscript at line 268-270 as follows:
“The “unknown” results were investigated manually and found to align accurately with reference sequences of known rare ABO alleles, indicating they were true variants rather than assay artefacts.”
Comments 4: This method is designed to identify known alleles. For most samples, this is an efficient and effective approach. However, as noted in the introduction, there are over 300 known alleles of the ABO gene. It is possible, and perhaps likely, that there are other, yet unknown, alleles that will be discovered as the method is employed at a larger scale in a clinical environment. I would suggest adding functionality to the pipeline, perhaps hidden from the user under normal circumstances, that will identify when a novel allele is found and bring that to the user’s attention so that it may be added to the database. This is especially important when the novel variant is an indel which would result in a frameshift mutation and what would otherwise appear to be an A or B allele resulting in an O phenotype. This should at least be mentioned in the limitations and future directions section.
Response 4: The current pipeline version identifies all single nucleotide variants (SNVs) in ABO exons 6 and 7 and provides Variant Call Format (VCF) files for each sample as part of the outputs to the user. However, it only classifies common ABO blood types based on the SNV positions listed in Table 3. Future versions aim to include distinguishing SNVs for A1 and A2 subtypes, which are particularly important for ABO-incompatible kidney transplants. This improvement is being pursued through a collaborative effort supported by the global nf-core bioinformatics community, with updates openly available at the community repository (https://nf-co.re/abotyper/dev/). There are no planned improvements to include any other additional rare subtypes outside the scope of this study. However, open sourcing this project ensures wider usage and development with the hope that this will extend this tool’s functionality.
Samples that don’t fulfill the distinguishing conditions defined in Table 3 are classified as “unknown”. “Unknown” results are due to samples failing to meet the SNV combinations, or the allele balance thresholds for heterozygous positions documented in Table 3. “Unknown” results caused by allele balance ratios may be resolved through manual inspection of the output excel worksheet and relaxing the threshold if appropriate.
We have reinstated the Table 3 footnotes (lines 162-165), which were truncated after submission, and updated the “Limitations and Future Directions” section of the manuscript (lines 402-onwards) to indicate these.
Comment 5: Despite the samples being normalized prior to pooling, there is a wide variation in the number of reads obtained in each sample. Approximately 20% of samples have fewer than the set lower limit of 20 reads and others have as many as several thousand reads. Is this due to uneven amplification of the amplicons in each pool? Is there any correlation with number of reads and allele that might indicate an allele with a SNV in the primer binding site? Did the samples with low numbers yield good HLA genotyping data? This failure rate was mentioned, but there was no discussion of why it was so high or any indication of effort to improve it. It will need to be improved prior to clinical use.
Response 5: Input DNA being derived from buccal samples is not quantitated or normalised prior to PCR amplification. Therefore, the DNA concentration and quality varies greatly between samples, which contributes to a range of amplicon concentrations. In order to maintain high throughput testing efficiency it was decided not to normalise DNA concentrations prior to PCR. The read variation seen between samples is likely due to the competition during the multiplex-PCR amplification between HLA and ABO genes, and then sequencing competition between the ABO amplicons and the HLA amplicons of various lengths.
To account for the variation in amplicon representation between the ABO and HLA genes and to minimise the number of ABO failures due to insufficient read depth, libraries are sequenced for longer to achieve more data per sample.
The data collected from the 910 samples didn’t show any allele specific amplification issues. Our 910 sample data set shows that all ABO genotypes reported similar failure rates. Samples with low reads (<20 reads for ABO) typically also failed to successfully amplify HLA loci, especially DQB1 which is also present in the mastermix with the ABO primers and is of similar amplicon size.
The failure rate of the assay was high due to the DNA derived from buccal samples which had increased issues of DNA fragmentation and less high-quality long strands. Text has been added to manuscript at line 257-260.
The failure rate of the 910 unique samples with a read depth of >20 was 5.4%. The expansive data set includes repeat testing which skews the interpretation of the real failure rate of the assay.
We have also added a summary text of the sequencing results for the 910 unique samples at read depths of ≥30×, ≥20× and ≥10× at line 276-286.
Reviewer 2 Report
Comments and Suggestions for Authors
Dear Sir
I have read with attention and interest the contribution: Characterisation of the ABO blood group phenotyping using a third generation sequencing. by F.M. Mobegi and collaborators.
The study presents a new method for determining the ABO genotype from a buccal / salivary swab that can be performed in parallel with HLA typing. The proposed genotyping method is based on a third generation sequencing method. The authors also propose a redundant and validated in-house model for the interpretation of the results.
From a conceptual point of view, I just want to underline, as also highlighted by the authors, that the validation work of the test for the genotyping of the ABO system would have been much simpler and also more robust if it had been performed on blood samples. Another aspect that leaves me perplexed is the presence of samples with labelling errors.
For the rest, it is a well-conducted study and the results are interesting.
From a formal point of view I would change the title: "Prediction of the ABO blood group phenotypes using third generation sequencing" or "Characterisation of the ABO blood group genotyping using a third generation sequencing".
Author Response
|
Comments 1: From a conceptual point of view, I just want to underline, as also highlighted by the authors, that the validation work of the test for the genotyping of the ABO system would have been much simpler and also more robust if it had been performed on blood samples. Another aspect that leaves me perplexed is the presence of samples with labelling errors. |
|
Response 1: While blood samples are often considered the gold standard for DNA extraction, the primary aim of this feasibility study was to assess the viability of buccal swabs as a non-invasive alternative for joining Stem Cell Donors Australia registry. Buccal swabs offer several advantages, including ease of collection, cost-effectiveness. Our large real-case study has also shown that swabs have the ability to obtain reliable DNA yields and quality comparable to blood samples. This makes buccal swabs particularly suitable for large-scale registry recruitment exercises and situations where minimising participant discomfort is a priority. We have added a sentence at line 89-93 to clarify the purpose of this study. Unfortunately, pre-analytical technical errors occur, and laboratories are constantly reviewing processes to ensure these errors are mitigated. Two samples showed discrepant result attributed to pre-analytical human/technical error. We have revised the text at line 235-243 to make this clear.
|
|
Comments 2: From a formal point of view I would change the title: "Prediction of the ABO blood group phenotypes using third generation sequencing" or "Characterisation of the ABO blood group genotyping using a third-generation sequencing" |
|
Response 2: We appreciate the reviewer’s suggestion regarding the title. After careful consideration, we have chosen to retain the original title, “Characterisation of the ABO blood group phenotypes using third-generation sequencing” as it accurately reflects the study's primary objective: to explore and define the capabilities of third-generation sequencing in classifying the ABO blood group phenotypes and underscores the methodological and analytical aspects of the research. “Prediction”, on the other hand, would shift the focus towards outcome estimation and fail to capture the original scope and intent of our study. |
Reviewer 3 Report
Comments and Suggestions for Authors
Summary
The paper describe a streamlined workflow for high-resolution ABO blood-group genotyping using Oxford Nanopore long-read sequencing. By appending a single 6.7 kb amplicon covering exons 2–7 to an existing HLA panel, they show that a handful of key variants can reliably distinguish A, B, AB and O subtypes. Their Nextflow‐based, containerized pipeline automates basecalling, alignment, variant calling and phenotype assignment. In 130 samples with at least 20× coverage, results agreed perfectly with serology and an external laboratory. Applied to 910 buccal-swab specimens, the assay passed quality filters in 95% of runs and resolved all ambiguous calls after targeted review.
Strengths
Comprehensive variant detection (SNVs and indels) across the entire target region using a single long amplicon
Fully reproducible, containerized analysis that yields identical outputs in test and production environments
Extensive validation against gold-standard serology, an independent lab and a large real-world cohort
Recommendations
Report sensitivity, specificity and predictive values, and include a confusion matrix to clarify performance for each blood group
Justify the 20× coverage threshold by showing accuracy trends at lower and higher depths
Describe any error-correction or polishing steps used to counteract nanopore basecalling errors, especially in homopolymer tracts
Provide per-base coverage plots across the amplicon to identify drop-out regions and guide any primer redesign
Explain how low-coverage or “unknown” calls are handled, including criteria for repeat PCR or manual review
Detail the demographic makeup of the validation cohort and plan testing in a genetically distinct population
Include hands-on and end-to-end processing times (lab and compute) to help labs plan resources
Discuss expanding the variant panel to capture rare alleles (e.g., cis-AB, A-subtypes) for broader clinical utility
Publish a user guide and open-source pipeline code under a permissive license to facilitate adoption
Compare per-sample costs (reagents plus cloud compute) with traditional serology and short-read genotyping
Conclusions
This work offers a robust, scalable solution for molecular ABO typing that combines the strengths of long-read sequencing with automated, reproducible analysis. Before moving toward clinical implementation, the authors should supply detailed accuracy metrics, cover error-correction and coverage analyses, validate in diverse populations, and provide user-friendly documentation.
Author Response
|
Comments 1: Report sensitivity, specificity and predictive values, and include a confusion matrix to clarify performance for each blood group. |
|
Response 1: We have included a confusion matrix and performance metrics between TGS and serology results for 861 unique samples with ≥20× read depth (Supplementary Table S3). These metrics are referenced in-text at line 270-276. |
|
Comments 2: Justify the 20× coverage threshold by showing accuracy trends at lower and higher depths. |
|
Response 2: Using a lower cut-off (≥10 reads) was also viable producing 100% concordant results but requiring more (3) manual assessments as compared to the higher threshold of ≥20×. We previously validated the 20× threshold for HLA genotyping assays (HLA 2024 Vol. 104 Issue 4 Pages e15725). Since this study aimed to assessing both HLA and ABO in a combined assay, a threshold of ≥20× reads was set. Using this threshold, 861/910 (94.6%) samples achieved 100% final concordance with serology and HLA genotyping, with only 1 sample requiring manual assessment. This empirically supported the 20× threshold as the minimum coverage for confident genotyping of both loci. We have added a summary of concordance at read depths of ≥30×, ≥20× and ≥10× at line 276-279 to text. |
Comments 3: Describe any error-correction or polishing steps used to counteract nanopore basecalling errors, especially in homopolymer tracts.
Response 3: Considering read-error correction comes at a substantial cost of compute resources and adds delay to the assay. The choice to include such a step must also outweigh the potential gains.
We designed our targeted amplicon sequencing workflow for ABO classification with multiple layers of error mitigation to ensure accurate results. To enhance homopolymer calling, we leveraged Dorado v0.5.3, a GPU-accelerated basecaller optimized for R10.4.1 chemistry. After basecalling, we used NanoFilt to filter out low-quality (<Q7) and short (<2000 bp) reads, ensuring only the most reliable data is retained for downstream analyses.
To minimize alignment-related errors, we trimmed primers from read ends to remove soft-clipping artifacts and improve downstream variant calling accuracy. We aligned the reads using minimap2, taking advantage of ONT-specific optimizations (e.g. -ax map-ont) for precise mapping. Our variant-calling pipeline relied on bcftools mpileup, applying stringent filters such as read depth (DP), base quality ≥ Q20, mapping quality ≥ Q20, allele frequency (AF), and allele balance to eliminate false positives and ensure biallelic support.
Our classification approach focuses on well-documented SNVs and small indels, including the widely studied 261delG in exon 6 for the O allele (doi: 10.1182/bloodadvances.2022007133). During assay development, we manually inspected alignments and variants using IGV (Integrative Genomics Viewer) to confirm consistency and eliminate sequencing artifacts. The pipeline outputs these alignments, variants, and associated metrics, allowing users to verify results or troubleshoot if needed.
We intentionally did not apply global homopolymer correction tools (e.g., Medaka, Homopolish) because our targeted assay design and high coverage effectively mitigate ONT error rates. Literature on ABO genotyping highlights that SNVs and short indels are the primary focus, rather than large homopolymer stretches. One reference explicitly mentions that " The main limitation is associated with about 1% error rate in the detection of changes during the sequencing of homopolymer fragments" in the context of NGS for blood group genotyping (Orzińska et al., 2021), but still achieved high accuracy, reinforcing that homopolymer-associated errors do not impact classification reliability.
With these design choices and bioinformatics controls, we are confident that our workflow provides accurate and robust ABO classification.
Comments 4: Provide per-base coverage plots across the amplicon to identify drop-out regions and guide any primer redesign.
Response 4: The pipeline generates per-base coverage metrics for exon 6 and exon 7 of each sample using SAMtools and BEDtools. This data is presented to users as part of standard outputs as mentioned in text [lines 191-194, updated line 204-207, updated methods line 335-338, and line 343]. These reports are further collated using MultiQC to provide a single per-run HTML output for convenience in tracking assay performance (See Figure 4). Adding plotting functionality for the BEDtools and SAMtools processes will only duplicate effort and increase the overall compute runtime.
Primers used for this assay have been validated for specificity and amplification intensity, and resulting sequencing data confirmed in over 1000 sequenced samples.
Comment 5: Explain how low-coverage or “unknown” calls are handled, including criteria for repeat PCR or manual review.
Response 5: The following sentence has been added at line 348-351 to clarify this:
Samples with <20× at relevant positions are flagged as "Low coverage" or "Unknown" in the “ABO_result.xlsx “, which is the main output of this workflow. These samples are excluded from reporting in the “final_export.csv” file and flagged for repeat testing or manual analysis if appropriate..
Comment 6: Detail the demographic makeup of the validation cohort and plan testing in a genetically distinct population
Response 6: The validation cohort includes samples from the Stem Cell Donors Australia Registry, with known enrichment of European, South Asian, and East Asian ancestry. A sentence has been added at line 100-102 to clarify the cohort demographic makeup.
At this stage, there are no plans to expand testing to genetically distinct populations outside the intended donor pool, as the primary objective of assessing feasibility for registry typing has already been met.
Comment 7: Include hands-on and end-to-end processing times (lab and compute) to help labs plan resources
Response 7: DNA extraction and PCR ~2 hours
Library Preparation ~ 3 hours
Sequencing time (MinION/GridION): ~16-24 hours
Bioinformatics pipeline (base calling to report): ~2-4 hours/batch run of 24 samples on Azure GPU/CPU compute.
This information has been added to the text at line 302-303 and line 346-347.
We recognise that these metrics will vary by user and available computing environments. The pipeline outputs a detailed HTML file, which summarizes for each sample all individual processes and resources used. Additional Nextflow-specific logs are also generated as standard outputs for each run to help with resources planning and allocation.
Comment 8: Discuss expanding the variant panel to capture rare alleles (e.g., cis-AB, A-subtypes) for broader clinical utility.
Response 8: The current variant panel is designed to detect four major ABO phenotypes and key O subtypes, tailored specifically for phenotype prediction in registry typing using buccal swabs from the target donor population. While the assay fulfills its intended purpose, we recognize the importance of broader clinical applicability. To support this, the project has been made publicly available and open to independent contributions via the nf-core bioinformatics community (https://nf-co.re/abotyper/dev/), ensuring transparency, collaboration, and accessibility.
The inclusion of additional rare alleles will be guided by user-driven needs and use cases. Any additions will undergo rigorous testing and validation and must be approved through the nf-core community to maintain the reliability and clinical robustness of the pipeline
Comment 9: Publish a user guide and open-source pipeline code under a permissive license to facilitate adoption.
Response 9: The full pipeline, along with installation instructions, is open-sourced and available on GitHub (https://github.com/fmobegi/abo-analysis) and the nf-core platform (https://nf-co.re/abotyper/dev). The project is released under an MIT open-source license for the code and a Creative Commons license for the documentation, enabling reuse and modification by the broader community.
Comprehensive documentation is provided, including details on core dependencies, setup instructions, example input/output files, and a deployment guide compatible with Singularity, Docker, and Conda environments.
Comment 10: Compare per-sample costs (reagents plus cloud compute) with traditional serology and short-read genotyping.
Response 10: This assay was not to replace serology but as a screening assay for high throughput registry typing. Costs may vary by country or setting, and we therefore chose not to include these estimates in the manuscript. Additionally, in our workflow, ABO typing is performed alongside HLA genotyping, so costs reflect the combined analysis for the purpose of registry typing only.